# Assessment of Optimism in Women with Polycystic Ovary Syndrome: A Case Control-Study

**DOI:** 10.3390/ijerph18052352

**Published:** 2021-02-28

**Authors:** Inés Morán-Sánchez, Evdochia Adoamnei, María L. Sánchez-Ferrer, María T. Prieto-Sánchez, Julián J. Arense-Gonzalo, Ana Carmona-Barnosi, Ana I. Hernandez-Peñalver, Jaime Mendiola, Alberto M. Torres-Cantero

**Affiliations:** 1Cartagena Mental Health Centre, Murcia Health Service, C/Real, 8, Cartagena, 30201 Murcia, Spain; ines.moran@carm.es; 2Department of Public Health Sciences, Division of Preventive Medicine and Public Health, Institute for Biomedical Research of Murcia, IMIB-Arrixaca, University of Murcia School of Medicine, 30100 Murcia, Spain; evdochia.adoamnei@um.es (E.A.); jaime.mendiola@um.es (J.M.); amtorres@um.es (A.M.T.-C.); 3Department of Obstetrics and Gynecology, Institute for Biomedical Research of Murcia, IMIB-Arrixaca, ‘Virgen de la Arrixaca’ University Clinical Hospital, El Palmar, 30120 Murcia, Spain; marisasanchez@um.es; 4Department of Public Health Sciences, Institute for Biomedical Research of Murcia, IMIB-Arrixaca, University of Murcia School of Medicine, 30100 Murcia, Spain; julianjesus.arense@um.es; 5Department of Obstetrics and Gynecology, ‘Virgen de la Arrixaca’ University Clinical Hospital, El Palmar, 30120 Murcia, Spain; ashanti770@hotmail.com (A.C.-B.); hernandezpenalver@gmail.com (A.I.H.-P.)

**Keywords:** polycystic ovarian syndrome, optimism, psychological factors, mental health

## Abstract

Polycystic ovary syndrome (PCOS) is a chronic endocrinopathy characterized by hyperandrogenism and anovulation that may pervade psychological dimensions such as dispositional optimism. Considering how PCOS influences mental health and the lack of studies on this matter, this research was aimed at assessing optimism and associated factors in PCOS. A case–control study of 156 patients with PCOS and 117 controls was conducted. All woman filled out the Life Orientation Test-Revised (LOT-R), a self-report questionnaire for measuring dispositional optimism. Medication, pain severity, gynecological, and sociodemographic information was also collected. Lower optimism was found in patients with PCOS compared to controls, even after covariate adjustment (LOT-R global scores: 14.1 vs. 15.9, *p* = 0.020). Our study provides evidence that a personality characteristic with important implications in illness prognosis may be affected in PCOS. We propose to assess dispositional optimism with the LOT-R scale in the gynecological appointment and tailor medical attention accordingly as a way to improve the comprehensive care of these patients within a multidisciplinary team.

## 1. Introduction

Polycystic ovary syndrome (PCOS) is a chronic endocrinological condition affecting 4–21% of women during their reproductive years, depending on the diagnostic criteria [1]. Patients show a broad range of symptoms for which hyperandrogenism and anovulation are responsible, such as hirsutism, menstrual and fertility disorders, obesity, and even emotional symptoms, which are extremely challenging for women diagnosed with this disorder [2]. Women with PCOS face the challenge of this multidimensional condition and must learn to cope with their situation. Their motivation and ability to implement and maintain successful lifestyle changes that are critical in this disorder may be affected by multiple factors [3], including patient’s individual predispositions such as dispositional optimism [2,4].

Optimism is a stable personality characteristic that reflects the extent to which people hold generalized favorable expectations for the future [4,5]. This personality dimension is associated with taking proactive measures to protect health [6] and with better treatment outcomes, quality of life, and lower risk of psychopathology [7,8,9,10]. This personality trait is resistant to downturn by adversity [11,12], but variations in optimism, both moment-to-moment and over extended periods, could also exist [7]. According to the theory of self-regulation, expectations of positive outcomes may decrease when people are faced with a succession of major adversities [13,14,15]. The belief that illnesses can be alleviated through one’s own actions may be hard to maintain when faced with a chronic illness that may be perceived as uncontrollable. For example, in a study which compared 110 patients with chronic fatigue disease and 47 healthy controls, authors found lowest optimism scores in participants with idiopathic chronic fatigue [16].

Most research on PCOS has been focused on physiological and diagnostic dimensions [17,18], but few studies have taken into account emotional aspects [19,20,21]. According to the literature, the challenges of body image and feminine identity due to acne, hirsutism, and obesity, as well as fertility problems and long-term health related concerns, may adversely impact on mood and psychological wellbeing [3,21,22]. All these factors together with the uncertainty about the course of the illness may affect their emotional state and the way these women face the future. Women with PCOS may feel they have little control over their illness and are more likely to have a pessimistic explanatory style. However, it would be interesting to know which was first, the disease or the altered personality trait, and thus establish the direction of causality [23]. We may argue whether PCOS itself is the cause or the consequence of lower optimism or whether some aspects of the disease are associated with mental suffering and may modify some personality traits.

Optimism has been studied in many medical conditions [7,24] but very little research has focused on gynecological diseases [2,25,26]. Only one previous study has been conducted on PCOS and there was no control group to compare with [2]. Therefore, the objective of this study was to assess dispositional optimism and associated factors in south-eastern Spanish women with PCOS compared to controls.

## 2. Materials and Methods

### 2.1. Participants

This case–control study was conducted between September 2014 and May 2016 at a University Clinical Hospital in southeastern Spain. The study design and method have been previously described [27]. Cases (*n* = 117) were women of 18–40 years of age diagnosed with PCOS attending the Gynecology Unit of the hospital. Cases were included only if a diagnosis could be established following the Rotterdam criteria, including medical history with Ferriman–Galwey scale [28], reproductive hormone levels, and transvaginal ultrasound. A diagnosis of PCOS required the presence of at least two of the following three criteria: (i) hyperandrogenism, either biochemical (total testosterone level ≥ 2.6 nmol/L) or clinical (mF-G score ≥ 12) [29] with or without acne or androgenic alopecia; (ii) oligo- and/or anovulation (menstrual cycles > 35 days or amenorrhea > 3 months); (iii) polycystic ovarian morphology (POM) on Transvaginal Ultrasound (TVUS) (≥12 follicles measuring 2–9 mm in diameter, mean of both ovaries) [30].

Controls (*n* = 156) were women without PCOS attending the gynecological outpatient clinic for routine gynecological examinations. Women were excluded from the study if they were pregnant or lactating, on oncological treatment, had any endocrine or gynecological disorders such as endometriosis or genitourinary prolapse, or were taking any hormonal medication, during the previous 3 months. More than ninety-five percent of the approached women agreed to participate in the study. Those that refused to participate did so mainly due to a lack of time for filling out questionnaires.

Written informed consent was obtained from all patients. The study was approved by the Ethics Research Committee of the Clinical University and the Hospital University of Murcia (no. 770/2013, approved 3 October 2013).

### 2.2. Procedure

For both groups, gynecologists recruited consecutive women attending the clinic. All participants completed health questionnaires, clinical, gynecological, and sociodemographic information at a scheduled visit. Variables related to optimism according to previous studies were also collected (education, professional situation, infertility, pain severity, and treatment for emotional disorders) [2,5,7,25,31]. Two gynecologists performed all clinical evaluations using the same methodology.

### 2.3. Materials

Weight and height were measured using a digital scale (Tanita SC 330-S, Amsterdam, The Netherlands), and BMI was calculated. Pain severity was evaluated using the 12-item Short Form Health Survey version 2(SF-12v2) bodily pain subscale, a 0–100 scale, with higher scores representing higher levels of health-related quality of life (HRQoL) [32]. This test is a validated shorter version of the SF-36v2 that includes 12 questions, which assess functional health and wellbeing from the participant’s point of view [33]. The SF-12v2 is a valid instrument for measuring HRQoL in our environment and has been previously used in Spanish populations [32,34].

Individual differences in generalized optimism versus pessimism were measured using the Life Orientation Test-Revised (LOT-R), a widely accepted test used in assessing dispositional optimism [35]. This test is a 10-item validated self-report instrument with three items referred to optimism, three items referred to pessimism, and the remaining four items being filler questions included in order to disguise the underlying purpose of the test. Respondents were asked to rate on a five-point Likert scale the extent to which each item was true or false. Total score is calculated by adding the inverted pessimism score and the optimism score, and ranges between 0 and 24 points with higher scores denoting more optimism. The results indicated that the properties of the Spanish version of the LOT-R were similar to those of the original test, and internal consistency reliability was acceptable for the total scale [36].

### 2.4. Statistical Analyses

Analyses were conducted with IBM-SPSS (version 25). Prior to analysis, all data were examined for normality and homogeneity of variance. Descriptive statistics were presented using raw data. Differences between groups on continuous data were analyzed using unpaired Student T tests, and group differences in categorical variables were compared with Pearson’s χ^2^ tests. With the analysis of covariance (ANCOVA), adjusted differences in LOT-R scores between cases and controls were assessed. Effect sizes were calculated using Cohen’s *d*, relating the mean score differences to the pooled standard deviation. Age, employment status, education level, infertility/sterility problems and bodily pain were included as potential confounders/covariates in the model in order to obtain a better adjustment. All tests were two-tailed at 0.05 significance level. From previous publications, it was considered that it would be appropriate to detect a difference of at least 1.5 points (with a standard deviation of about 4.3 points) in the LOT global score between cases and controls for the calculation of the simple size. Accepting a risk of 0.05 and 80% of statistical power, should there be any differences, 113 subjects would be needed in one group and 150 in the other.

## 3. Results

Sociodemographic and clinical characteristics of participants are described in Table 1. Cases were younger than controls (27.2 ± 5.0 vs. 30.7 ± 5.9 years, *p* < 0.001). Among women with PCOS, infertility/sterility problems were more prevalent (PCOS: 24 (20.5%); controls: 13 (8.4%), *p* < 0.001). Women with PCOS also had higher levels of bodily pain than controls—lower scores in the SF12v2 questionnaire— (81.3 ± 24.6 vs. 90.5 ± 18.3 points, *p* < 0.001). There were also significant differences between cases and controls regarding employment status or educational level (*p* < 0.001).

Raw and adjusted LOT-R scores for women with PCOS and controls are shown in Table 2 and Figure 1. LOT-R optimism mean score was significantly higher in controls (8.3 vs. 7.7 points; *p* = 0.03) compared to cases. On the other hand, cases obtained higher LOT-R pessimism score than controls (5.6 *vs.* 4.3; *p* < 0.01), even after the adjustment (*p* = 0.022). Finally, controls obtained an LOT-R global score of 1.8 points higher than PCOS women (*p* < 0.01) and this relationship persisted after adjustment (*p* = 0.020), which indicates significantly higher levels of optimism in controls. The effect size of this difference was relatively high (*d* = 0.50).

## 4. Discussion

We believe this study is important because it is the first one to specifically assess dispositional optimism on PCOS women with a case–control design. In this sample of Spanish women in reproductive age, lower optimism was found in women with PCOS compared to controls. Optimism is one of the resource-oriented dimensions of increasing importance in the last few years in psychological research. There is a current trend of identifying new ways to complement the traditionally deficit-oriented approach in clinical and health psychology [5]. This research might contribute to the analysis of other relevant aspects of emotional wellbeing in patients with PCOS beyond depression or anxiety syndromes [19,20].

The available data on optimism among women suffering from gynecological diseases is scarce. Previous investigations have assessed optimism in the context of PCOS, endometriosis, and in vitro fertilization treatments [2,25,26]. These authors found that response to in vitro fertilization treatments may be hindered by personality characteristics [25]. Very little research has focused on gynecological diseases using standardized instruments. Our group have recently assessed optimism in a sample of 95 Spanish women with endometriosis using the same questionnaire [26]. Women with endometriosis obtained an LOT-R global score of 14.5 ± 3.9 points, 1.4 points lower than controls, even after full adjustment (*p* = 0.045), which indicates lower levels of optimism in patients diagnosed with this condition. Regarding the measurement of optimism in PCOS using the LOT-R questionnaire, only one previous study applied this test in women with PCOS, but there was no control group to compare with [2]. This study found LOT-R global scores of 13.56 ± 4.28 points in 250 women diagnosed with PCOS, being findings quite similar to those obtained by our group. Another quantitative study [5] applied the LOT-R questionnaire in the general population and found LOT-R optimism scores of 8.8 ± 2.5 points, LOT-R pessimism scores of 4.5 ± 2.4 points, and LOT-R global scores of 16.2 ± 3.8 points. In our study, controls obtained similar global and optimism scores but higher pessimism scores (7.6 vs. 4.5 points).

Why can optimism be affected in PCOS? PCOS affects body image and feminine identity due to obesity, hirsutism, and acne. Besides, pervading concerns about crucial aspects of a woman’s life, such as fertility and long-term health-related concerns are common in patients diagnosed with PCOS [3,22]. Women with PCOS may feel they have little control over their illness and therefore, they may be more likely to have a less optimistic style. Moreover, a severe chronic condition such as PCOS can affect cognitive perceptions and coping strategies, such as the optimistic explanatory style. All these factors together with the uncertainty about the course of the illness may affect their emotional state and the way these women see the future.

Another reason that may affect dispositional optimism is pain. Growing evidence suggests that, although personality traits are considered relatively stable, people may alter their personality profiles, for instance, after developing ongoing pain [23,37,38]. In this study, we found that woman with PCOS have higher bodily pain severity than controls as measured by SF12v2. As pain is not one of the characteristic symptoms of PCOS, the fact that patients suffered from more severe bodily pain does not necessarily mean that it is caused by the disease, it may be also multidetermined. Previous research concluded that patients with painful chronic gynecological conditions were more likely to experience pessimistic worries [37]. There is evidence that in people with chronic pain, some personality traits may be altered through psychological therapy [39]. If a personality trait can be influenced through therapy in people with chronic pain, it may be possible to conclude that, conversely, an altered personality trait might, in fact, have been developed as a consequence of the pain itself [40]. In this research, we considered factors such as infertility problems or pain severity that may affect optimism and even so, dispositional optimism was lower in PCOS. The burden of a chronic disease characterized by pervading concerns may affect this psychological dimension [4], as has been reported for other chronic diseases, such as endometriosis, chronic fatigue syndrome, multiple sclerosis, or diabetes mellitus [16,26,41]. However, it would be interesting to know which was first, the condition or the altered personality dimension, that is, “the egg or chicken causality dilemma”, and thus establish the direction of causality [23].

These findings may have significant research and clinical implications. Given that the LOT-R questionnaire is a fast, low participant burden and reliable tool that may assess optimism in an easy and specific way, we propose to consider incorporating it into daily gynecological practice. Furthermore, as optimism is related to psychopathology risk and response to treatments [23], we consider that PCOS would require an integrated approach combining medical and psychological interventions such as personality assessment [21]. It might also be very useful to involve other health providers such as clinical psychologists in the treatment of PCOS [42]. This multidisciplinary team must have developed a specific expertise on PCOS for a better management. In this sense, the approach to every woman with PCOS should be individualized, taking into account her preferences and proposing the best treatment options based on the different symptoms, the problems distressing the woman, or her reproductive desires [37,40]. In order to improve wellbeing and health outcomes, these findings suggest that we need to attribute more importance and effort to therapeutic strategies aimed at psychological traits such as optimism. One possible way of improving treatment effectiveness is to include trait-focused psychological therapy to adequately deal with all the multifaceted repercussions often experienced by patients with PCOS [23]. It opens up possibilities for improving emotional state, patient satisfaction and not developing future psychopathology. Future studies are needed to evaluate the effectiveness of these targeted psychotherapies.

There are several limitations that should be considered. Firstly, controls belonged to the same population of women with PCOS and they attended the hospital during the same period. A second limitation that should be considered arises from the cross-sectional data origin. There is no information on optimism before the diagnosis of PCOS, so it is hard to determinate the direction of causality. It would be useful to collect longitudinal data on this condition in order to better document variations in this important trait over time. Another limitation is that we did assess bodily pain and we did not collect different conditions that may influence this pain.

## 5. Conclusions

PCOS is a multidimensional condition with a broad range of symptoms, including emotional ones, that may affect many aspects of a patient’s life. This study concludes that a personality trait with many prognostic implications may be altered in women with PCOS. We propose to evaluate optimism in a simple way in the gynecology visit and, accordingly, adapt the medical attention to these patients as a way to improve the comprehensive care within a multidisciplinary team. Our findings highlight the importance of a broader understanding of PCOS, treating this condition from a biopsychosocial perspective.

## Figures and Tables

**Figure 1 ijerph-18-02352-f001:**
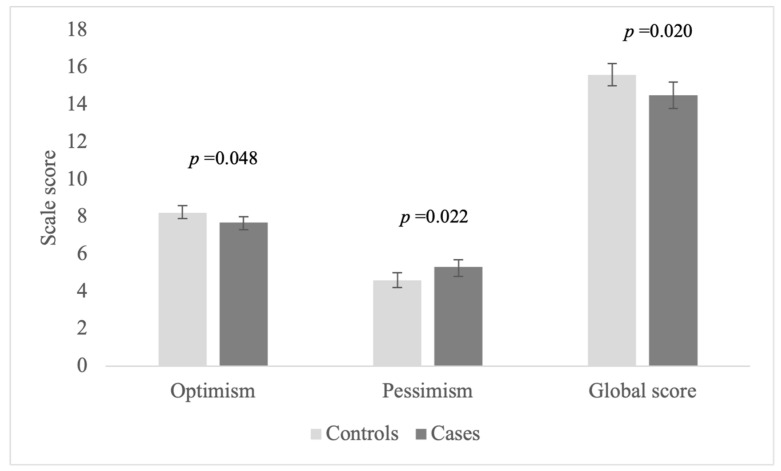
Adjusted means LOT-R optimism, LOT-R pessimism, and LOT-R global scores and 95% confidence intervals in cases of polycystic ovary syndrome and controls (*p* = 0.020).

**Table 1 ijerph-18-02352-t001:** Characteristics of patients with polycystic ovary syndrome and controls.

Variable	Cases (*n* = 117)	Controls (*n* = 156)	*p*-Value *
Age (years), (mean and SD)	27.2 (5.0)	30.7 (5.9)	<0.001
Body Mass Index (Kg/m^2^), (mean and SD)	25.5 (5.9)	23.4 (4.5)	0.001
Marital status, *n* (%)			
Single or divorced	56 (47.9)	76 (48.7)	0.89
Married	61 (52.1)	80 (51.3)	
Educational level, *n* (%)			
Primary	25 (21.9)	15 (9.7)	0.001
Secondary	38 (33.3)	39 (25.2)	
University	51 (44.7)	101 (65.2)	
Employment status, *n* (%)			
Unemployed	24 (21.2)	22 (14.1)	0.09
Studying	32 (28.3)	35 (22.4)	
Working	57 (50.4)	99 (63.5)	
Infertility/sterility problems, *n* (%)	24 (20.5)	13 (8.4)	<0.001
Bodily pain (mean and SD) **	81.3 (24.6)	90.5 (18.3)	<0.001
Psychiatric medication, *n* (%)			
Anxiolytics	8 (6.8)	9 (5.8)	0.73
Antidepressants	6 (5.1)	6 (3.9)	0.63

SD = Standard Deviation. * Student *t*-Test or Chi-squared (χ^2^). ** Bodily pain is a 0–100 subscale from the SF12v2 questionnaire, with higher scores representing higher levels of health-related quality of life.

**Table 2 ijerph-18-02352-t002:** Comparison of the Life Orientation Test-Revised (LOT-R) between cases of polycystic ovary syndrome and controls.

LOT-R Variables	Cases (*n* = 117)	Controls (*n* = 156)	*p*-Value ^a^	Adjusted *p*-Value
	Mean (SD)	Mean (SD)		
LOT optimism	7.7 (2.3)	8.3 (1.9)	0.03	0.048 ^b^
LOT pessimism	5.6 (2.6)	4.3 (2.2)	<0.01	0.022 ^c^
LOT global score	14.1 (3.9)	15.9 (3.2)	<0.01	0.020 ^d^

SD = Standard Deviation. ^a^ Unadjusted value (Student *t*-Test). ^b^ Adjusted by age and employment status. ^c^ Adjusted by age, educational level, and infertility/sterility problems. ^d^ Adjusted by age, educational level, employment status, and infertility/sterility problems.

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
