# Peer review of "Assessment of Optimism in Women with Polycystic Ovary Syndrome: A Case Control-Study"

_ijerph, 2021, doi:10.3390/ijerph18052352_

Round 1

Reviewer 1 Report

A very good manuscript. In the opinion of the reviewer, the only remark is the suggestion to change the first sentence. The authors did not avoid many stylistic errors. After their removal, the work can be accepted for printing

Author Response

REVIEWER #1:

Comments to the Author

A very good manuscript. In the opinion of the reviewer, the only remark is the suggestion to change the first sentence. The authors did not avoid many stylistic errors. After their removal, the work can be accepted for printing

Response: We thank you for your comments. We have changed the first sentence and a native English speaker has revised the manuscript.

Reviewer 2 Report

This is a well-prepared manuscript. The methods applied in this study are correct. Findings are interesting and novel.

There are two minor comments:

  1. Can the authors explain why cases were younger than controls? Any hypothesis? 
  2. This study was carried out between 2014 and 2016. Do the authors think, that the year of the study up to 6 years ago can influence the obtained findings?

Author Response

REVIEWER #2:

Comments to the Author

This is a well-prepared manuscript. The methods applied in this study are correct. Findings are interesting and novel.

There are two minor comments:

  1. Can the authors explain why cases were younger than controls? Any hypothesis? 

Response: Cases were woman of reproductive age diagnosed with PCOS. Controls were women without PCOS attending the gynaecological outpatient clinic for routine gynaecological examinations. We might hypothesize that the usual symptoms associated with PCOS (e.g., oligomenorrhea, hyperandrogenism, etc.) may lead to an early detection of the condition (younger age) and the profile of women attending the outpatient clinic for routine exams may be a little bit older because these examinations are conducted in different periods of a woman's life including post-menopause.

2. This study was carried out between 2014 and 2016. Do the authors think, that the year of the study up to 6 years ago can influence the obtained findings?

Response: We agree with the reviewer that there is a delay from data collection to publication, but we do not believe this delay may have influenced the findings. To the best of our knowledge, there is only one published article about optimism and PCOS and data, also collected in 2016, are quite similar to ours [Rzonca et al]. 

  • Rzońca E, Iwanowicz-Palus G, Bień A, et al (2018) Generalized Self-Efficacy, Dispositional Optimism, and Illness Acceptance in Women with Polycystic Ovary Syndrome. Int J Environ Res Public Health 15:1–10. https://doi.org/10.3390/ijerph15112484

Reviewer 3 Report

Article of current interest.

Results are presented that would help to understand psychologically the states of women suffering from polycystic ovaries.

However, certain changes should be made in the presentation of the study:

1- Avoiding the use of the first person plural
2- To make the method and materials section clearer, I subdivided it into participants, procedure, materials and analysis
3- Use more formal and academic language, for example avoid saying "so" but "thus".
4- In general, review the language used to make it more scientific.

Author Response

REVIEWER #3:

Comments to the Author

Results are presented that would help to understand psychologically the states of women suffering from polycystic ovaries.

However, certain changes should be made in the presentation of the study:

1- Avoiding the use of the first person plural
2- To make the method and materials section clearer, I subdivided it into participants, procedure, materials and analysis

3- Use more formal and academic language, for example avoid saying "so" but "thus". 
4- In general, review the language used to make it more scientific

Response: Following reviewer suggestions, a native english has reviewed the language and we have subdivided methods and materials sections as follows:

  1. Materials and Methods

2.1. Participants

2.2 Procedures

2.3. Materials

2.4. Statistical Analyses